# Overexpression of *AtruLEA1* from *Acer truncatum* Bunge Enhanced *Arabidopsis* Drought and Salt Tolerance by Improving ROS-Scavenging Capability

**DOI:** 10.3390/plants14010117

**Published:** 2025-01-03

**Authors:** Shaofeng Li, Huijing Meng, Yanfei Yang, Jinna Zhao, Yongxiu Xia, Shaoli Wang, Fei Wang, Guangshun Zheng, Jianbo Li

**Affiliations:** 1State Key Laboratory of Tree Genetics and Breeding, Experimental Center of Forestry in North China, National Permanent Scientific Research Base for Warm Temperate Zone Forestry of Jiulong Mountain in Beijing, Chinese Academy of Forestry, Beijing 100091, China; lisf@caf.ac.cn (S.L.); yxxia@caf.ac.cn (Y.X.); wshaoli@iccas.ac.cn (S.W.); wangfei@caf.ac.cn (F.W.); 2College of Forestry, Henan Agricultural University, Zhengzhou 450046, China; 15938277261@163.com; 3College of Forestry, Shanxi Agricultural University, Taigu 030801, China; 13759225665@163.com (Y.Y.); jinnazz@163.com (J.Z.)

**Keywords:** *Acer truncatum* Bunge, *LEA1*, drought stress, salt stress, ABA

## Abstract

*Late embryonic developmental abundant* (*LEA*) genes play a crucial role in the response to abiotic stress and are important target genes for research on plant stress tolerance mechanisms. *Acer truncatum* Bunge is a promising candidate tree species for investigating the tolerance mechanism of woody plants against abiotic stress. In our previous study, *AtruLEA1* was identified as being associated with seed drought tolerance. In this study, *LEA1* was cloned from *A. truncatum* Bunge and functionally characterized. *AtruLEA1* encodes an LEA protein and is located in the nucleus. Phylogenetic tree analysis revealed a recent affinity of the AtruLEA1 protein to AT3G15760.1. Overexpression of *AtruLEA1* resulted in enhanced tolerance of *Arabidopsis thaliana* to drought and salt stress and heightened the ABA sensitivity. Compared to wild-type (WT) plants, plants with overexpressed *AtruLEA1* exhibited increased activities of antioxidant enzymes under drought stress. Meanwhile, the ROS level of transgenic *Arabidopsis* was significantly less than that of the WT. Additionally, the stoma density and stoma openness of *AtruLEA1 Arabidopsis* were higher compared to those in the WT *Arabidopsis* under salt and drought stress conditions, which ensures that the biomass and relative water content of transgenic *Arabidopsis* are significantly better than those of the WT. These results indicated that *AtruLEA1* was involved in salt and drought stress tolerances by maintaining ROS homeostasis, and its expression was positively regulated by abiotic stress. These results indicate a positive role of AtruLEA1 in drought and salt stress and provide theoretical evidence in the direction of cultivating resistant plants.

## 1. Introduction

Plants may be exposed to various types of abiotic stress, like water stress and high and low temperatures, which can significantly hinder their normal growth and development [1]. To cope with environmental stress, plants have evolved various strategies, such as increasing the root length, increasing the activity of antioxidant enzymes and inducing the expression of antioxidant genes, for example, late embryonic developmental proteins [2,3,4].

LEA proteins are a diverse family of proteins which are mainly located in the nucleus, cytoplasm and mitochondria [5]. They play an important role in protecting cells under abiotic stress [6]. As a large number of large family proteins, LEA proteins are usually small and hydrophilic molecules that contain. High proportions of Gly, Ala, Glu, Lys/Arg and Thr, but low or absent Cys and Trp.

LEA proteins have a major role in shielding cells from abiotic stresses, including cold, drought and salinity, which could induce their production [7,8,9,10]. The transgenic Arabidopsis with RcLEA from Rosa chinensis showed better growth and less peroxide after high- and low-temperature treatment [11]. The analysis of the expression profile showed that most LEA genes were highly expressed in drought-tolerant varieties compared with drought-sensitive upland cotton [12], which suggests that the LEA protein was related to drought stress. The LEA family gene (PtrLEA7) was isolated from Poncirus trifoliata, and it enhanced the drought resistance to some extent by improving the antioxidant capacity [13]. The MSLEA4-4 gene from Medicago sativa L. improved the drought resistance of Arabidopsis by improving the activity of antioxidase [14]. AtLEA3-3 improved the drought tolerance and extended the vegetative growth cycle of Arabidopsis [15]. The research and exploration of LEA genes in plants is beneficial for the cultivation of drought-resistant varieties and the study of drought-resistance mechanisms, such as in banana, which helped with the cultivation of drought-tolerant banana varieties [16]. These results also showed that it was of great significance to understand the response mechanism of plants to abiotic stress and improve plant stress resistance by studying LEA genes.

LEA proteins have been divided into several categories according to their amino acid sequence characteristics [17]. These proteins were conserved hydrophilic proteins, which participated in various adaptive responses to hypertonic conditions. The expression of the LEA gene was usually ABA-dependent, not only in seeds but also in water-deficient vegetative tissues, related to drought, salt and cold stress. All identified LEA genes were grouped into seven groups, ranging from LEA 1 to LEA 6 and seed maturation protein (SMP). There were large differences between different groups. The Group 1 LEA proteins (Pfam PF00477), originally from the cotton seed D-19 and D-132 proteins, have a very large proportion of charged residues that contribute to the high hydrophilicity and high content of Gly residues (about 18%) [18]. The Group 2 LEA proteins (Pfam PF00257), also known as “dehydrins”, were initially identified as the D-11 family in cotton embryos. This group of LEA proteins is among the best-characterized groups. In terms of Group 2 LEA, the protein is characterized by a conserved 15-residue motif K fragment (EKKGIMDKIKEKLPG) enriched in Lys [19]. In addition, the histone also contains a set of S fragments, which is rich in serine, and its phosphorylation has some effect on subcellular localization detection after phosphorylation [20]. In addition, in the process of Arabidopsis research, it was found that only S fragments can be phosphorylated, but the effect of phosphorylation is different. The phosphorylation of S fragments in Arabidopsis can affect the activity of bound calcium ions [21]. The Group 3 LEA proteins (Pfam PF02987) are characterized by a repeat motif of 11 amino acids. The molecular weight differences found in this set of proteins are usually the result of the different copy numbers of this 11-mer motif. This histone has a conserved domain (ΦΦ E/QX Φ KE/QK Φ XE/D/Q) in which Φ is a hydrophobic amino acid. This fragment allows multiple repeats and is present in animals and plants as well as in prokaryotes [22]. The Group 4 LEA proteins (Pfam PF03760) are widespread in the plant kingdom, including non-vascular plants and vascular plants. The D-113 protein from cotton was the first discovery in this group. This histone N terminus has α-helix binding and domain conservation, but the C terminus is poorly conserved [23]. They represent the non-canonical LEA subfamily with no obvious conserved domains because they contain a higher proportion of hydrophobic residues [24].The third and fourth groups are induced under dehydration, which could maintain the 3D structure of cells and the glass state of cells. Among them, the third group may have some connection with ABA induction [25]. The wheat TaEm protein (LEA family 5) could improve the tolerance to high salt in *S. cerevisiae* [26].

Reviewing the literature revealed that the fifth group of LEA proteins has low identity and poor stability, and the protein is insoluble in water in a boiling water bath. This histone is also induced by various abiotic stresses, and studies have found that a low temperature, drought, UV light and mechanical damage can induce the expression of the LEA proteins in this group [9,23]. Boucher [27] found that MtPM25 can form hydrogen bonds to prevent proteolytic folding due to water stress, which can age condensed proteins after eliminating water stress. The group 6 LEA proteins have a small molecular weight (about 7–14 kD) and are highly conservative. The structure of domain one (LEDYKMQGYGTQGHQQPKPGRG) is conserved with domain two (GSTDAPTLSGGAV). Currently, 36 genes in this family have been found in vascular plants. It is under the common bean roots and the cotyl elongation or accumulation of growth areas, making its water potential lower than the nongrowth zone with enhanced water absorption capacity. Soybean-related studies show that the protein is distributed in the apical meristem, root tip, and vascular tract sites, which are all visible. The transport of plant water and nutrients and root tissue development occur. Seven groups of LEA proteins are considered the smallest, heat-stable high proteins among the LEA family members [28]. The groups of genes have been identified in a variety of dicots and monocots as well as in gymnosperms. However, in Arabidopsis, no ASR analogue genes were found. They share physiological and chemical properties with other LEA proteins in the embryo. They accumulate in seeds in late fetal development as well as under water shortage.

*Acer truncatum* is a tree species of great importance, possessing not only medical and economic benefits but also significant ecological value; moreover, research has previously shown that this organism has a high level of stress resistance [29]. Specifically, transcriptomic data were analyzed to further understand the underlying mechanisms, and it was found that late embryogenesis abundant (LEA) proteins are responsive to stress and are associated with the species’ native environment [30,31]. Previous research has also shown that AtruLEA1 levels are very high under abiotic stress; these raw RNA-seq data for *A. truncatum* under drought conditions have been deposited in the National Genomics Data Center database (PRJCA013039) (https://bigd.big.ac.cn, accessed on 13 November 2022) [30]. To further explore the stress resistance mechanism of this gene, in this study, it was overexpressed in Arabidopsis. The structure of AtruLEA1 was found to contain an alpha helix, which can improve the ability to retain water. The promoter element of 1156 bp upstream of ATG was analyzed and found to contain cis-acting elements that react with ABA and homeotypic elements for auxin synthesis, which is closely related to drought and salt stress. By analyzing the expression patterns of AtruLEA1, it was found that its overexpression increased tolerance to salinity and drought stress by improving ROS scavenging in transgenic Arabidopsis. Furthermore, AtruLEA1 improved the sensitiveness, which helps plants to sense the emergence of stress as soon as possible and enter the defense mode as soon as possible. In summary, the findings of this research study provide a molecular basis for delving deeper into the stress resistance mechanism of *A. truncatum*, as well as a fresh perspective for stress resistance breeding.

## 2. Results

### 2.1. Isolation and Characterization of AtruLEA1

The coding sequence (CDS) of *AtruLEA1* (Atru.chr8.189) measures 546 bp in length. The sequences of late embryogenesis abundant (LEA) proteins from *Arabidopsis* were sourced from the TAIR database, and a phylogenetic tree was generated by utilizing the neighbor-joining (NJ) method, as illustrated in Figure 1A. The AtruLEA1 protein was observed to be closely related to a significant clade of AT3G15670.1, which is classified within the superfamily of proteins categorized into several subgroups exhibiting highly hydrophilic structures. Conserved motifs were identified by using the online tool MEME, revealing that AtruLEA1 proteins, which share functionally analogous roles, displayed comparable motif compositions and conserved sequences (Figure 1B).

### 2.2. Protein Domain Prediction

The online prediction tool provided by the National Center for Biotechnology Information (NCBI) indicates that the AtruLEA1 protein is classified within the superfamily PTZ00121, specifically as a member of the cl3157 family. This protein exhibits multiple structural domains, and two LEA domains can be seen (Figure 2B). Analyses conducted by using SMART and PFAM revealed that AtruLEA1 encompasses structural domains characteristic of the LEA4 family proteins (refer to Figure 2A,B). Furthermore, this protein was also found to possess the MA, Dak2 and HABP4-PAI-RBP 1 domains (see Figure 2C).

### 2.3. The Structure of AtruLEA1

The prediction of subcellular localization indicates that the AtruLEA1protein may reside in the nucleus, similarly to the PtrLEA7 protein [9]. We used SOPMA (https://npsa.lyon.inserm.fr/cgi-bin/npsa_automat.pl?page=/NPSA/npsa_sopma.html accessed on 26 April 2024) and SWISS-MODEL (https://swissmodel.expasy.org/) software to predict the secondary and tertiary structures of AtruLEA1, where the former predominantly consists of α helices, random coils and β folds, as illustrated in (Figure 3A,B).

### 2.4. Analysis of AtruLEA1’s Physicochemical Properties

The physicochemical properties of molecular weight and pI value of the AtruLEA1 protein were analyzed by using ProtParam software (https://web.expasy.org/protparam/, accessed on 1 January 2024). The results are shown in Table 1.

### 2.5. Analysis of AtruLEA1’s Signal Peptide and Protein Transmembrane

The analysis of the transmembrane structure of the protein showed that AtruLEA1 has a high probability of being outside the membrane, a low transmembrane probability and no transmembrane structure, thus being classified as a non-transmembrane protein (Figure 4A). SignalP 4.0 online software was used to predict the signal peptide of the AtruLEA1, showing that it does not exist in the protein sequence; thus, AtruLEA1 is a non-secreted protein (Figure 4B).

### 2.6. Analysis of AtruLEA1 Promoter

Plant Care online software (http://bioinformatics.psb.ugent.be/webtools/plantcare/html/, accessed on 1 January 2024) was used to analyze and predict the number of cis-acting elements contained in the upstream 1156 bp promoter sequence of the AtruLEA1 gene (Figure 5). We found a large number of cis-acting elements, including anaerobic response inducer (ari), ABRE (an element involved in anaerobic acid responsiveness), ACE (an element related to photo-responsiveness), AuxRR core (an element involved in auxin reaction), ARE (an element involved in anaerobic induction), Box 4 (part of the conserved DNA module involved in photo responsiveness), G box (an element involved in light responsiveness), TCA element (an element involved in the salicylic acid response), MYB binding sites (MBSs) and TATA box. Because this study focused on abiotic stress, we aimed to identify the abiotic-stress-related functions of AtruLEA1.

### 2.7. AtruLEA1 Improved the Drought and Salt Resistance in Arabidopsis

The *LEA* gene was confirmed to be involved in various abiotic stress types, and previous transcriptomic data analysis in this experiment showed that *AtruLEA1* is significantly upregulated under drought stress. To further investigate the function of *AtruLEA1*, it was overexpressed in *Arabidopsis*. Five independent T2 overexpression transgenic lines were obtained with qRT-PCR (Figure 6A). Among the OE plants, OE-3 and OE-10 presented significantly higher relative expression of *AtruLEA1* than the other transgenic lines; therefore, they were selected for further drought and salt tolerance studies. The WT and two homozygous *AtruLEA1* lines (OE-3 and OE-10) were cultured for one week and then transferred to 1/2 MS medium containing mannitol and NaCl for two weeks (Figure 6B).

There was no difference in growth between WT and *AtruLEA1* seedlings under normal conditions. However, the *AtruLEA1* seedlings displayed increased resistance when subjected to mannitol and salt stress compared with the WT. When grown on mannitol, the WT seedlings had a length of 3.14–3.5 cm and a fresh weight of 3–4 mg, whereas the *AtruLEA1* seedlings were 4.8–5.8 cm in length and 0.0053–0.008 g in fresh weight. Under salt treatment, the growth of the *Arabidopsis* seedlings was significantly inhibited, but this phenomenon was relatively mild in terms of indicators. The main root length of the WT seedlings was 4.1–4.5 cm, and the fresh weight was 0.008–0.016 g, while root lengths of 4.9–5.3 cm and fresh weights of 0.0101–0.015 g were recorded for both OE-3 and OE-10. The number of lateral roots in the transgenic group was higher than that in the WT under every condition (Figure 6C,D).

The stress tolerance capacity of transgenic plants overexpressing the *AtruLEA1* gene was assayed under greenhouse conditions. Under drought and salt stress, the growth of wild-type plants was highly suppressed (Figure 7A,B). Following treatment, the leaves of the WT seedlings wilted severely under water deficit conditions, while all transgenic plants remained tender and green. Furthermore, white bleaching occurred in *Arabidopsis* leaves under salt stress conditions, and the bleaching rate of the transgenic plants was lower than that of wild-type *Arabidopsis*. The survival rate of the transgenic plants after drought stress treatment was 100%, while that of the WT was 0%. Under salt stress, the survival rate of transgenic *Arabidopsis* was 60 to 80% and that of the WT was 40% (Figure 7C). The relative water content (RWC) of the *AtruLEA1* transgenic plants was significantly higher than that of the WT plants, which was approximately 30% to 40% under normal conditions, but 20% under salt stress. The relative water content of transgenic *Arabidopsis* under drought conditions was 40%, and that of the WT was 0% (Figure 7D). In addition, under salt and drought stress, the relative electrical conductivity (REC) of the *AtruLEA1* transgenic plants was lower, i.e., 30% to 45%, than that of the WT, which was 50% to 60% (Figure 7E).

### 2.8. Overexpression of AtruLEA1 Enhances Sensitivity of Transgenic Plants to ABA

To further investigate the resistance molecule mechanism of AtruLEA1 and the relationship between stress and ABA response, we determined the expression levels of related genes, specifically two ABA biosynthesis-related genes (NCED9 and NCED5), under normal and drought conditions after four hours of drought stress (Figure 8D,E). Further, we measured the root length and fresh weight of WT and transgenic seeds (Figure 8B,C). Under normal conditions, there was no significant difference in the expression levels of NCED5 and NCED9 between the WT and the transgenic plants. Under drought conditions, the expression of the ABA synthesis-related genes (NCED9 and NCED5) in the transgenic plants was two to three times higher than that in the WT plants. The fresh weight and main root length showed no difference under control conditions. Under 50 μM ABA treatment, the fresh weight and root length of the WT were significantly better than those of the transgenic plants. These results suggest that the overexpression of *AtruLEA1* increased the expression of genes related to ABA synthesis and sensitivity to ABA (Figure 8A). Therefore, *AtruLEA1* could be involved in abiotic stress resistance through the ABA-dependent pathway.

### 2.9. Effect of Drought and Salt Stress on Stomata in Arabidopsis

Stomatal density and size have important effects on plant growth and development. Therefore, these characteristics were observed in the transgenic and wild-type plants in different growth environments. Under normal growth conditions, stomatal density was higher in the transgenic specimens than that in the wild type. Changes in the size and number of stomata in *Arabidopsis* were recorded under drought and salt stress. Under drought stress, the density of stomata significantly increased, but their opening decreased. The increase in stomatal density, which ensures plant growth and development on the premise of reducing water loss, was higher in transgenic *Arabidopsis* than in the wild type. Under salt stress, on the other hand, there was no significant change in stomatal density, and most of the stomata tended to close, but in transgenic *Arabidopsis*, the phenomenon of the stomata tending to close was slightly less evident (Figure 9A,B).

### 2.10. AtruLEA1 Mediates ROS Scavenging Capability

Abiotic stress can result in the production of ROS, including H_2_O_2_ and O^2−^. To assess the amount of ROS, the levels of these two species were determined based on 3,3′-diaminobenzidine (DAB) and nitro blue tetrazole (NBT) staining, respectively. Under normal growth conditions, no significant difference was observed in DAB and NBT staining between *AtruLEA1* transgenic plants and WT plants. However, after salt and drought treatment, the staining signal in WT plants was darker than that in the *AtruLEA1* seedlings (Figure 10A,B), indicating lower levels of hydrogen peroxide and O^2−^ in the *AtruLEA1* plants under stress.

To explore the synergistic relationship between the *LEA* gene and antioxidant enzymes, in this study, the fresh leaves of wild-type and transgenic plants in the same habitat were used to determine the levels of antioxidant enzyme activity. Under normal conditions, there were no significant differences in the activities of peroxidase (POD), superoxide dismutase (SOD) or catalase (CAT) enzymes between transgenic lines and wild-type plants. After drought and salt stress treatment, POD, SOD and CAT activity in the transgenic lines was higher than in the wild-type plants, indicating that *AtruLEA1* improved *Arabidopsis’* drought and salt stress resistance. Therefore, we believe that the overexpression of *LEA* is positively correlated with the activity of several antioxidant enzyme systems (Figure 10C–E).

## 3. Discussion

LEA proteins exist in a wide range of plant species, with more than 30 species having been identified so far, and are associated with various physiological and bio-chemical reactions in plants [6]. These proteins are classified into eight groups, and most of them are hydrophilic and can bind to water [16]. The amino acid number of LEA proteins is relatively small, generally ranging from 90 to 500 amino acids. For example, potato (*Solanum tuberosum*) LEA3 (AEJ 08687.1) contains 98 amino acids [32], cotton (*Gossypium hirsutum*) LEA1 (AFH57283.1) 165 amino acids [33], rice (*Oryza sativa* L.) LEA3 (AAD02421.1) 200 amino acids [34], LEA1 (A2XG55.2) 333 amino acids [34,35] and *Arabidopsis* LEA4 (AEC09277.1) 448 amino acids [6,16]. LEA proteins are mainly expressed in the stems and roots of plants and are less expressed in other organs [36]. As we also found in our study, group 2 LEA proteins can be detected in dried seeds and plants under drought, high-salt and low-temperature stress [37]. The heterologous expression of LEA proteins in several hosts, such as yeast, confers resistance to various stress types [38]. *A. truncatum* Bunge is an excellent example of resistant tree species, with strong drought and salt stress tolerance functions. In our study, we subjected it to drought stress and found that the *AtruLEA1* gene is drought-induced, which prompted us to select it as a drought alternative gene for further research.

Based on transcriptome data, *AtruLEA1* was identified as a candidate gene involved in drought tolerance in *A. truncatum* Bunge; thus, in this study, its function regulation in response to abiotic stress was investigated [16]. The phylogenetic analysis indicated that AtruLEA1 is closely related to At3G15670.1 (Figure 1) and belongs to a superfamily of proteins (Figure 2A) which are related to water stress [38]. Based on SMART and Pfam analyses, we found that AtruLEA1 has many protein domains, such as LEA4 domains, HABP 4_PAI-RBP1 domains, an MA domain and a Dak 2 domain (Figure 2C). AtruLEA1 was found to be a relatively short protein, measuring only 181 amino acids. At the N terminus, the AtruLEA1 protein sequence has a highly conserved α-helix, and at the C terminus, it has a low-complexity density region. The LEA-4 protein has an N-terminal α-helix that is known to be involved in signaling or transport in plant cells during water shortage and is able to adsorb water molecules, maintaining cell homeostasis [39]. This speculation was confirmed by the higher relative water content in *AtruLEA1*-overexpressing plants than in wild-type plants under normal and drought stress conditions (Figure 7D). The domain of this family, which includes the HABP4 hyaluronan-binding proteins and the PAI-1 mRNA-binding protein PAI-RBP1, is located between positions 4 and 73 of the protein. The MA domain, located between amino acids 5 and 158, is a chemotactic-like domain that is responsible for accepting methyl groups and transmitting signals to CheA and is highly conserved among various MCPs. The Dak2 domain, located between amino acids 24 and 139, is a phosphatase domain of the dihydroxyacetone kinase family, a family of enzymes which are capable of phosphorylating dihydroxyacetone, glyceraldehyde and other short-chain ketones and aldehydes, using ATP or phosphoenolpyruvate (PEP) as a source of high-energy phosphates [21,23]. The promoter analysis of the AtruLEA1 genes revealed that they contain ABA-binding elements and regulatory elements related to growth and development. (Figure 5). In summary, these structures can explain why the *AtruLEA1* improves the ability of *Arabidopsis* thaliana to resist against abiotic stress.

Large LEA protein accumulation is a resistance mechanism under osmotic stress. Other reports have shown that LEA overexpression in plants can improve plant tolerance to drought, salt, cold and heat stress conditions collectively [40] or to a single type of abiotic stress [41]. In this study, the function of *AtruLEA1* was assessed by overexpressing it in *Arabidopsis thaliana*. The transgenic plants showed superior growth ability under salt and drought conditions, as evidenced by increased root length and fresh weight compared with the WT (Figure 7). Abiotic stress causes plants to produce large amounts of reactive oxygen species (ROS), which damage cell membranes, cause lipid peroxidation and limit plant growth and gas exchange rates. In order to survive under drought stress, plants reduce material metabolism, close stomata and slow down gas exchange to ensure that the species reproduces [42]. Salt stress significantly increases Cl^−^ accumulation in plants, especially in leaves, and excessive Cl^−^ accumulation destroys the electron acceptors of the photosynthetic pigment and photoreaction system in the chloroplast; this is one of the reasons for the reduced gas exchange rate in plants and leaf stomata becoming long and narrow [43]. Salt stress suppresses plant growth by consuming energy for photosynthetic assimilation [44]. The stomatal changes in *Arabidopsis* in different growth environments observed in this study are consistent with the above findings (Figure 9).

In many studies, *LEA* was mainly induced by drought conditions, which may be related to cis-acting elements in its promoter region [45]. For example, ABRE and AuxRR in the promoter region of *AtruLEA1* may be involved in abscisic acid signaling pathways, which play a crucial role in plants exposed to stress conditions (Figure 5). Studies suggest that abscisic acid signaling pathways activate a series of stress-related genes under drought stress, which increases the survival probability of plants under adverse conditions [46]. ABA is essential to responding to changes in gene expression caused by water stress and plays an important role in regulating the expression of stress-induced genes [47]. Through various signal transduction pathways, ABA induces physiological changes in plants such as stomatal closure, proline synthesis and ROS clearance [48]. In this study, the seeds of *35S::AtruLEA1* were much more sensitive to ABA than those of the WT, and the expression of ABA synthesis-related genes was higher under drought stress (Figure 8).

ROS are generated following ABA treatment and abiotic stress, and the enzymatic defense system is responsible for eliminating excess ROS in plants. In this study, when *AtruLEA1*-overexpressng plants were exposed to salt and drought stress, the activity of POD, SOD and CAT was increased and the accumulation of ROS was reduced, indicating the positive role of *AtruLEA1* in ROS scavenging (Figure 10). The results of this study are consistent with the function of LEA proteins in plants according to other studies, which showed that these proteins increase the tolerance of plants to stress by boosting the activity of antioxidant enzymes [12,14,33]. In addition, the lack of water causes the leaves to wilt, which in turn leads to the collapse of the plant cell membrane, causing the electrolyte solution to disperse. The accumulation of LEA proteins, according to this study, contributes to the structural stability of various proteins in the cell and, thus, to maintaining the cellular water-holding capacity. We observed that under osmotic stress, conductivity in transgenic plants was lower than in wild-type *Arabidopsis*, while relative water content was higher (Figure 7D,E), which also proves the function of the LEA proteins. These results are consistent with previous studies [33,48]; the overexpression of *LEA* rapidly produces a large amount of LEA proteins in the plant, thus slowing down the wilting of plant cells due to water loss.

Lastly, we speculated that AtruLEA1 can jointly activate ABA synthesis genes and the activity of antioxidant enzymes to achieve plant regulation under abiotic stress. The overexpression of *AtruLEA1* significantly improves salt and drought tolerance in plants by reducing ROS levels and increasing the activity of antioxidant enzymes. The protein also has an α-helix at the N end of the amino acid sequence, which can adsorb water molecules and improve water retention in plants, thus significantly improving their survival rate [49,50]. The promoter contains abscisic acid-binding components and photo-response elements, which are important in stomatal opening and closing. Plants maintain growth by regulating the density, opening and closing of leaf stomata to ensure biomass accumulation. The present study provides a basis for achieving a deeper understanding of the function and molecular mechanisms of AtruLEA1 in *Arabidopsis thaliana* under salt and drought stress.

## 4. Materials and Methods

### 4.1. Plant Material and Treatments

*A*. *thaliana* (Col-0) was selected as a transgene receptor. First, *Arabidopsis* seeds were sterilized with 75% alcohol 3 times. Afterwards, they were dried and scattered evenly on 1/2 MS medium. For vernalization, the seeds were kept at 4 °C for 2–3 days and then cultured under the conditions of a temperature of 22–25 °C and a 16:8 light–dark ratio [39,49]. After they had germinated for one week, they were transferred to the soil.

### 4.2. Acquisition of Total RNA and cDNA

The analysis of transcriptome data showed that the gene is induced by drought and highly expressed in roots and seeds [34]. RNA was isolated by using an RNeasy Plant kit (Tiangen, Beijing, China), and cDNA was obtained with a Tian gen Biotech Fast King cDNA first and first-strand synthesis kit (TIANGEN). Their absorptive values were measured with a micro-spectrophotometer (Colibri, New York, NY, USA) to verify their quality.

### 4.3. Obtaining AtruLEA1 and Quantitative Real-Time PCR

The cDNA obtained was used as a template, and specific primers were designed with Primer 5 to amplify the full length of *AtruLEA1* from *A. truncatum* Bunge (Appendix A). Further, we obtained the primer for qRT-PCR by using primer 3, with *Actin1* as a reference, and gene expression was evaluated by using the 2^−ΔΔCt^ method with three biological replicates [50] (Appendix A).

### 4.4. Bioinformatic Analysis of AtruLEA1

*Arabidopsis* LEA family protein sequences were obtained from the *Arabidopsis* database, and a phylogenetic tree was constructed with MEGA 5.0. The molecular weight and isoelectric point of the AtruLEA1 protein were determined by using the ExPASy program (http://www.expasy.org/tools/pi_tool.html, accessed on 1 January 2024). The amino acid sequence motif of this protein was predicted by using the MEME online website. The 1156 bp upstream of the *AtruLEA1* start codon sequences was submitted to PlantCARE (http://bioinformatics.psb.ugent.be/webtools/plantcare/html/, accessed on 1 January 2024) for promoter cis-acting element prediction [51].

### 4.5. A. thaliana Transformation and Isolation of Transformed Plants

These experiments were based on the gene sequence and CDS of *AtruLEA1*. We cloned and constructed the latter into the overexpression vector pCAMBIA1302, driven by the 35S promoter, labeled *35S:: AtruLEA1*. The gene-containing plasmid was introduced into Agrobacterium. Moreover, *Arabidopsis* was transformed with the vector containing the target gene by means of *Agrobacterium*-mediated immersion [52]. The T0 seeds were screened on selection medium containing 25 mg/L hygromycin, and 5 transgenic lines were confirmed with qRT-PCR. Two independent *AtruLEA1* transgenic lines with high *AtruLEA1* expression levels were employed for the evaluation of stress tolerance [51,52,53]. The seeds from highly expressing homozygous T2 lines were selected for subsequent experiments (Appendix A).

### 4.6. Drought and Salt Stress Tolerance Analysis

During the seedling period, the seeds of *35S::AtruLEA1* and WT plants were cultivated on 1/2 MS medium for one week and then on 1/2 MS medium containing mannitol and NaCl for 10 days for the measurement of fresh weight and primary root length in the salt and drought tolerance experiments.

For the experiments in the adult seedling stages, a group of the three-week-old *AtruLEA1* transgenic *Arabidopsis* and WT seedlings grown in soil did not receive water for 10 days and were then rehydrated for three days. The other group was subjected to salt treatment with the administration of 50 mL of 200 mM NaCl through irrigation for two weeks. Afterwards, we assessed the plants’ growth ability and determined some physiological parameters of the seeds.

### 4.7. Sensitivity to ABA

One-week-old seedlings of *35S::AtruLEA1* and WT plants were cultivated on 1/2 MS agar plates, and they were then transferred to a medium with 50 µM ABA for sensitivity assessment based on the measurement of root length and fresh weight. Further, we determined the expression of ABA biosynthesis-related genes (NCED1 and NCED5) in *35S::AtruLEA1* and WT seedlings under drought stress to determine the relation between ABA and *AtruLEA1* under this condition [54] (Figure 9).

### 4.8. Observation of Stomata

To determine stomatal density, the samples were observed with an electron microscope, Motic, at a magnification of 40×. The number of stomata in three fields of view of each sample was counted by using Photoshop, averaged and divided by the area of the image to obtain the number of stomata on a 1 mm^2^ leaf blade (stomatal density (pcs/mm^2^)) [55]. In addition, stomatal size was determined by observing the sample with a 16 × 40 microscope; specifically, 10 fields of view were selected for each specimen, and each was measured once. Stomatal length was measured as the length of the kidney-shaped body in the stomatal apparatus, and stomatal width was determined as the widest value of the stomatal apparatus perpendicular to the kidney-shaped body [56].

### 4.9. Physiological Measurements and Histochemical Assays

Primary leaves of *35S:: AtruLEA1* and WT plants were selected for physiological assays, i.e., REC, RWC and enzyme activity, under normal, drought stress and salt stress conditions. To measure REC and RWC, we referred to previous methods [57]. An assay kit (Solarbio, Beijing, China) was used to determine the activity of antioxidant enzymes (SOD, POD and CAT). Each index measurement was repeated three times.

For DAB staining, a DAB staining solution at a 1 mg/mL concentration was adjusted to pH 3.8 with dilute hydrochloric acid. Leaves were placed in the DAB staining solution and vacuum-infiltrated with a vacuum pump for 20 min. They were then placed in the dark at 25 °C for 6–8 h [58]. For NBT staining, the NBT staining solution at a 0.5 mg/mL concentration was adjusted to pH 7.8. Leaves were placed in the NBT staining solution and vacuumed for 20 min; then, they were placed in the dark at 25 °C for 3 h [59]. After staining, the leaves were decolorized in a 95 °C water bath in 95% anhydrous ethanol. Lastly, images were taken by using a Zeiss Axio Imager A1 microscope (Carl Zeiss, Oberkochen, Germany).

### 4.10. Statistical Analyses

Statistical analyses were conducted by utilizing SPSS statistics 19.0 (SPSS Inc., Chicago, IL, USA). The data underwent a comparison process based on the *t*-test. When the *p*-value was below 0.05, the variances were regarded as significant, and highly significant differences were denoted when *p* < 0.01 (**).

## Figures and Tables

**Figure 1 plants-14-00117-f001:**
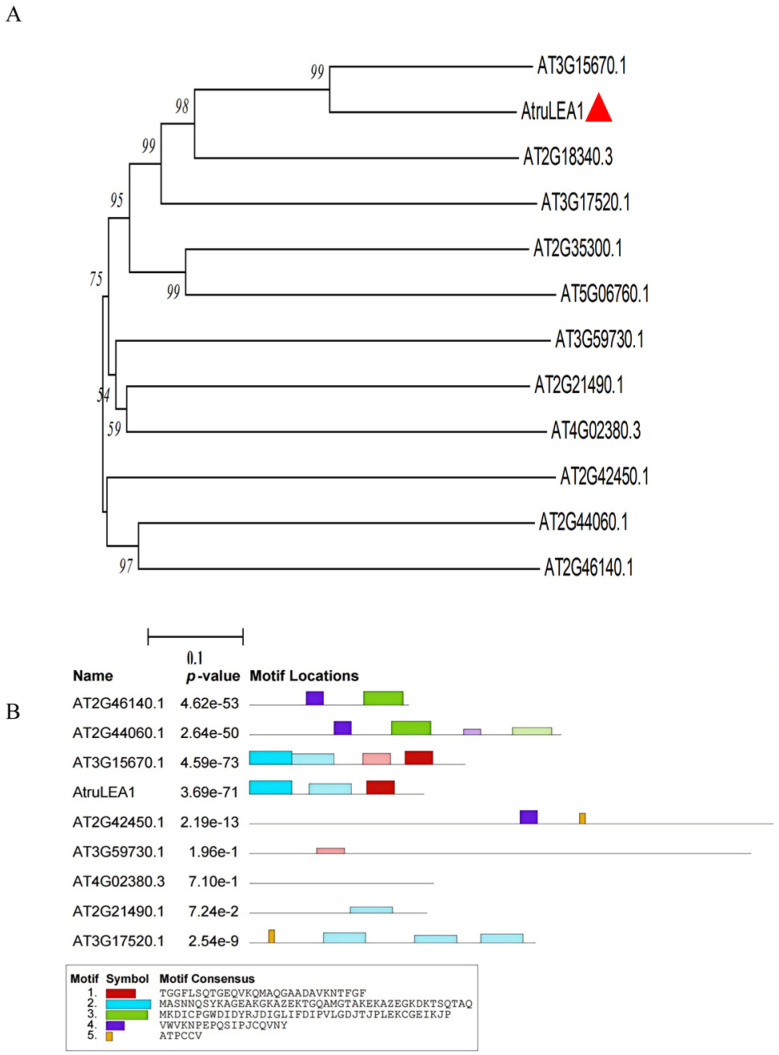
The phylogenetic analysis and conserved motif patterns of AtruLEA1. (**A**) The phylogenetic relationship between AtruLEA1 and LEAs from *Arabidopsis*. (**B**) Conserved motifs were identified by using the MEME tools. The five predicted motifs are represented by distinct colored boxes, and the gray lines indicate non-conserved regions for AtruLEA1.

**Figure 2 plants-14-00117-f002:**
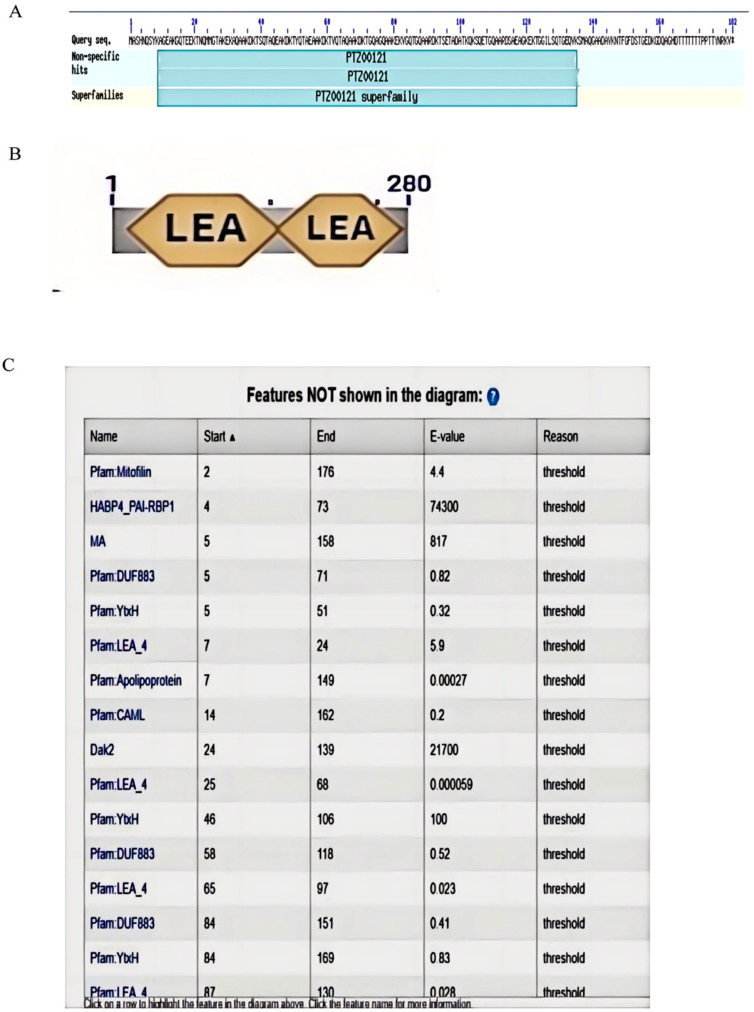
Protein domain prediction. (**A**) This image shows a graphical summary of the conserved domains identified on the query sequence. The domains are color-coded according to the superfamilies to which they have been assigned. Hits with scores that pass domain-specific thresholds (specific hits) are drawn in bright colors; others (non-specific hits) and superfamily placeholders are drawn in pastel colors. (**B**) The AtruLEA1 protein’s conserved domain. (**C**) The domain not shown in the diagram.

**Figure 3 plants-14-00117-f003:**
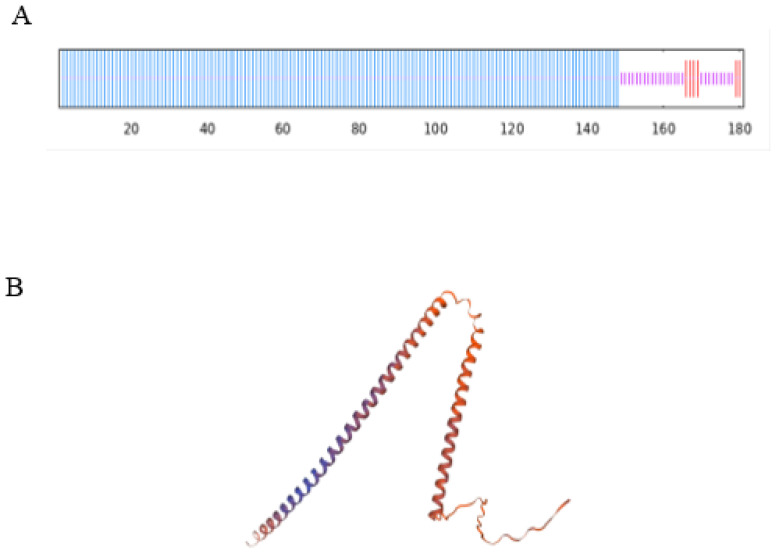
The protein structure analysis of AtruLEA1. (**A**,**B**) The secondary and tertiary structures of the AtruLEA1 protein.

**Figure 4 plants-14-00117-f004:**
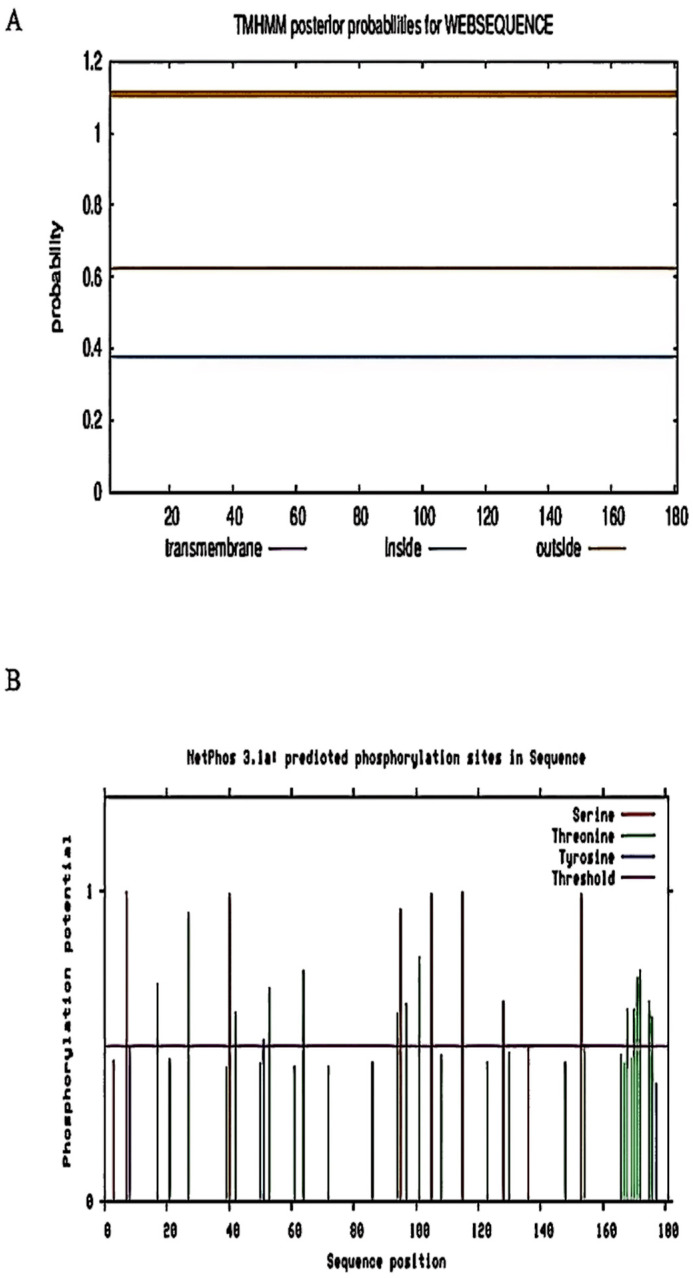
The transmembrane structure and the signal peptide. (**A**) AtruLEA1’s transmembrane structure. The inner part represents the intracellular region and the outer part the extracellular zone; the transmembrane represents the transmembrane region; and the larger the value of the transmembrane, the greater the possibility that this amino acid is in the transmembrane region. (**B**) The signal peptide: The abscissa is the protein sequence, while the ordinate is the probability.

**Figure 5 plants-14-00117-f005:**
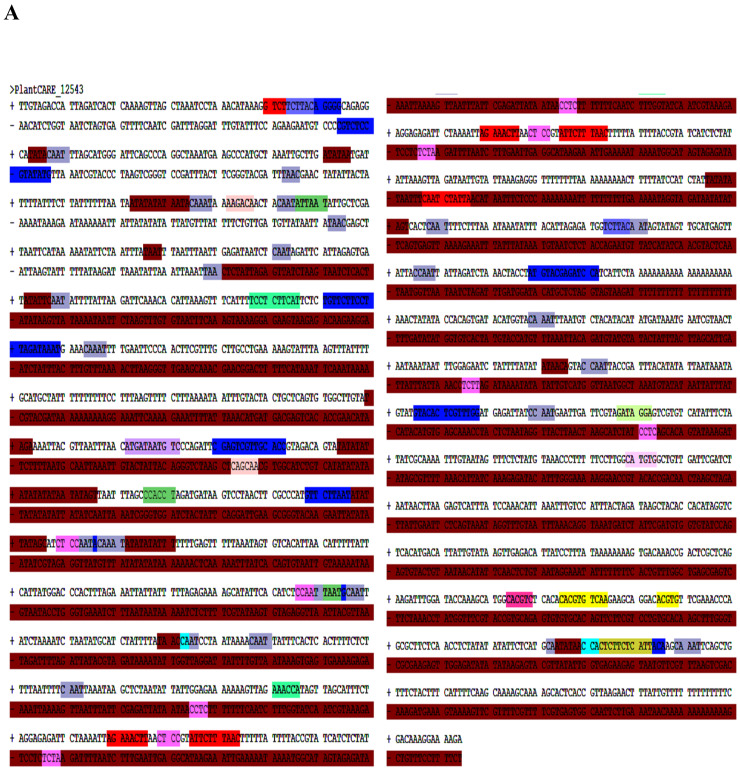
Bioinformatic analysis of AtruLEA1 promoter. The promoter sequence information of the genes was obtained from figShare websites and analyzed by using Plant Care online tools. Multiple drought-related cis-acting elements were found in the promoter region of AtruLEA1. (**A**) The sequence of the AtruLEA1 promoter and the cis-acting elements. (**B**) Different colors correspond to different cis-acting elements.

**Figure 6 plants-14-00117-f006:**
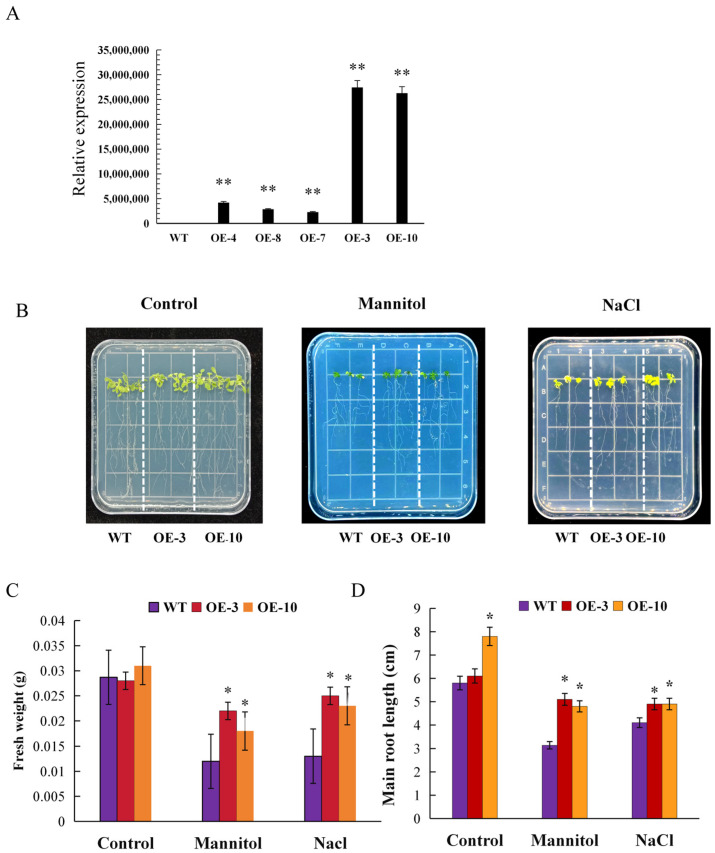
Drought and salt tolerance analyses of *AtruLEA1* transgenic seedlings under panel growth conditions. (**A**) The expression levels in *AtruLEA1* transgenic and WT seedlings. (**B**) An image showing *35S::AtruEA1* (OE-3 and OE-10) and WT seedlings under normal conditions, NaCl treatment and drought treatment in a plate. (**C**,**D**) The fresh weights and root lengths of both WT and AtruLEA1-overexpressing seedlings were measured under normal conditions and under salt and drought stress treatments. The significance of differences among the lines was tested by using Student’s *t*-test (* *p* < 0.05 and ** *p* < 0.01).

**Figure 7 plants-14-00117-f007:**
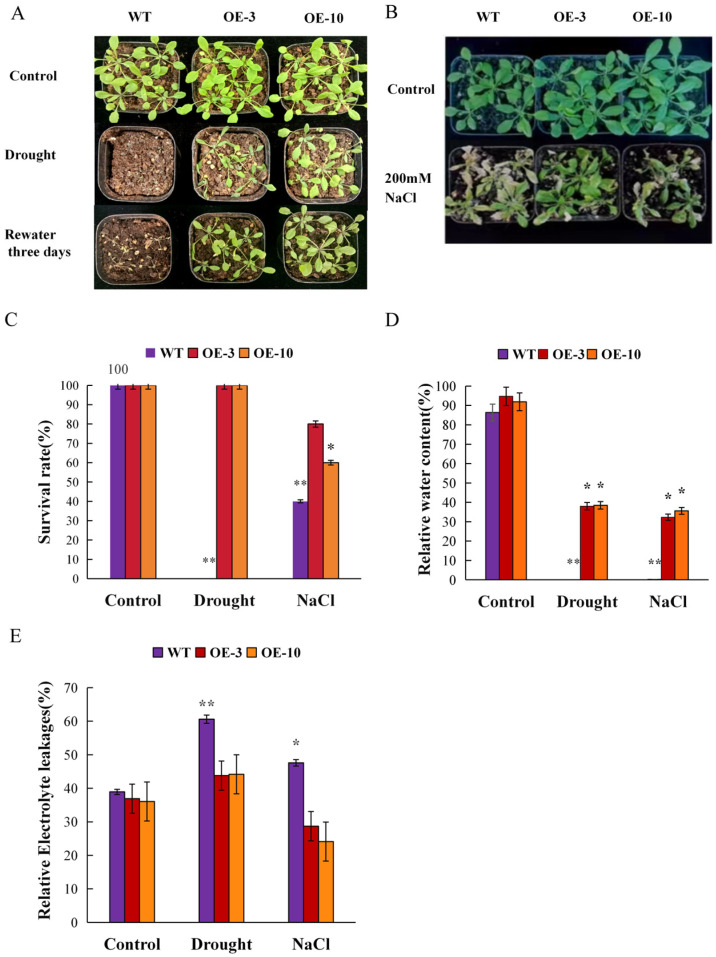
Transgenic *AtruLEA1 Arabidopsis* seedling resistance to salt and drought stress in soil. (**A**) *AtruLEA1*-overexpressing and WT seedlings were cultivated in soil under normal growth conditions for two weeks. Following water supply withholding for 10 days, we rehydrated the plants to examine their ability to resist drought stress. (**B**) For salt treatment, the seedlings were irrigated with 50 mL of 200 mM NaCl every 3 days for 2 weeks. Photographs were taken after each course of treatment. (**C**) The survival rates of WT and *AtruLEA1*-overexpressing seedlings under control, salt stress and drought stress conditions. (**D**,**E**) The relative water content (RWC) and relative electrical conductivity (REC) of WT and *AtruLEA1*-overexpressing seedlings under control, salt stress and drought stress conditions. The values presented herein are the result of an average calculation based on three separate replicates. Student’s *t*-test was used to determine significant differences (* *p* < 0.05 and ** *p* < 0.01).

**Figure 8 plants-14-00117-f008:**
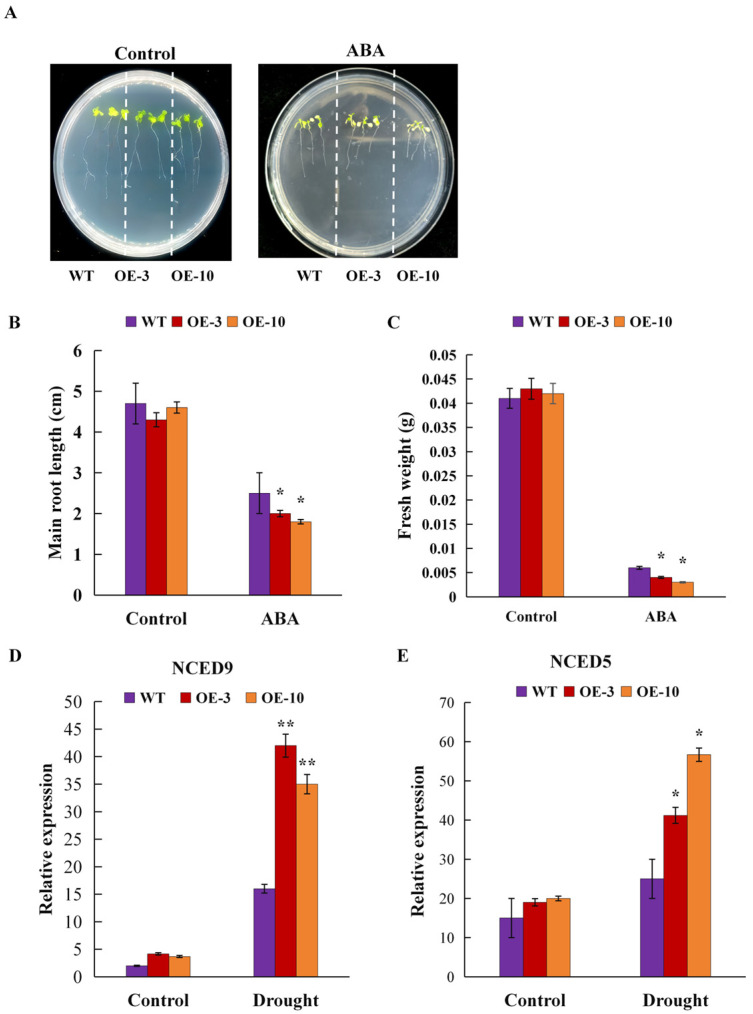
Early-seedling experiment under ABA treatment. (**A**) Seedlings in 1/2 MS solid medium and 1/2 MS solid medium with ABA (50 μM) for 5 days. (**B**,**C**) Effects of ABA treatment on root length and fresh weight. A photo was taken before measuring the root length and the fresh weight of the seedlings subjected to the different treatments. (**D**,**E**) The expression of the ABA synthesis-related genes (NCED9 and NCED5). The vertical bars represent the means ± standard deviations. Student’s *t*-test was used to determine significant differences (* *p* < 0.05 and ** *p* < 0.01).

**Figure 9 plants-14-00117-f009:**
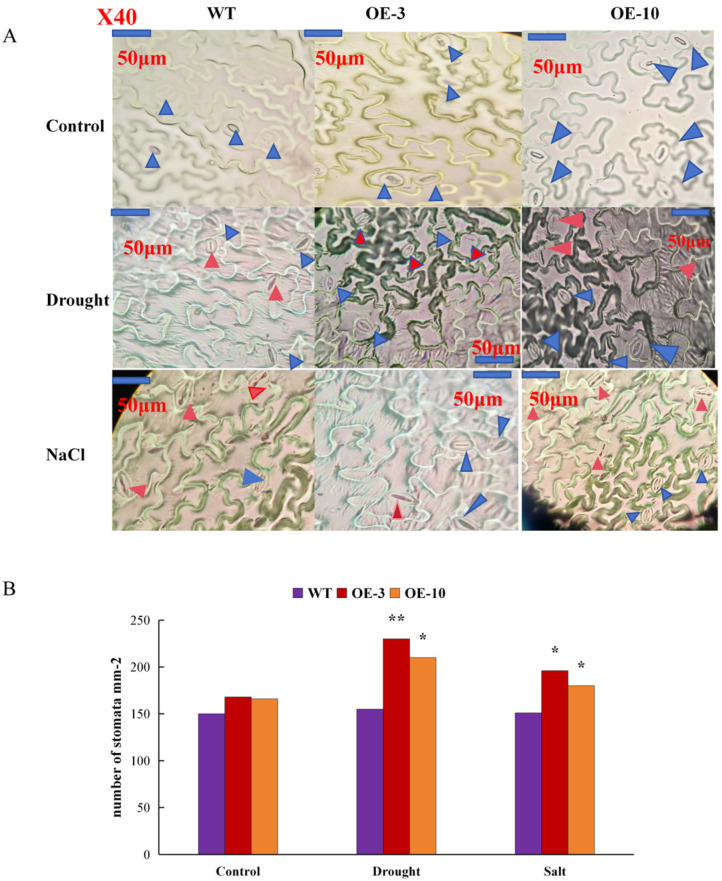
(**A**) Stomatal density in OE-10, OE-3 and wild-type plants under drought and salt stress conditions. (**B**) Stomatal density is shown as the average number of stomata per square millimeter. The number of 
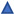
 and 
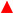
 means number of stomata in *Arabidopsis*, 
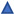
 means the normal stomata, 
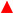
 means the stomata tended to close. All the images in the figure were acquired at 40× magnification under a microscope. Student’s *t*-test was used to determine significant differences (* *p* < 0.05 and ** *p* < 0.01).

**Figure 10 plants-14-00117-f010:**
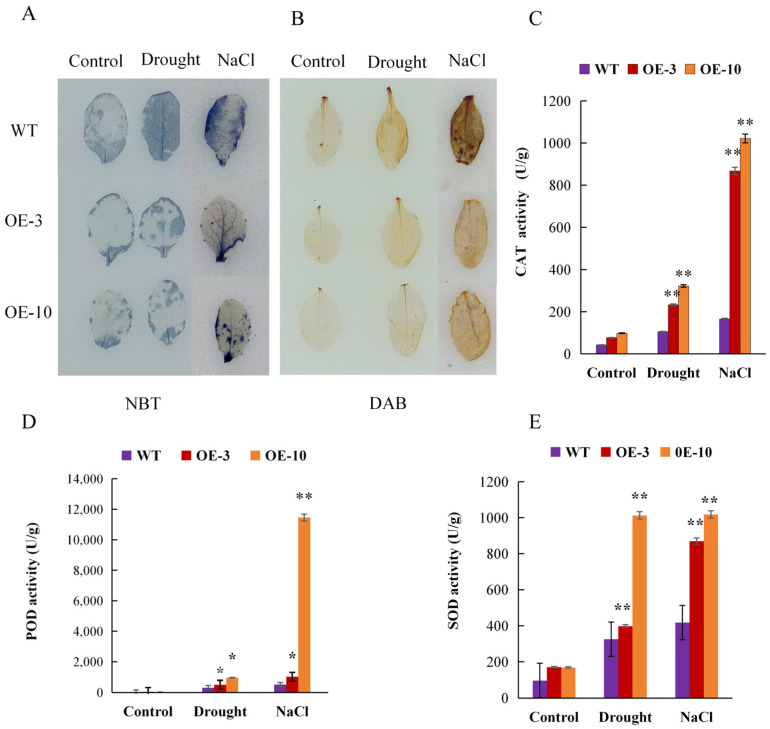
*AtruLEA1* mediates ROS scavenging ability. (**A**,**B**) The NBT and DAB staining of leaves from *35S::AtruLEA1* (OE-3 and OE-10) and the WT under normal and stress treatments. (**C**–**E**) CAT, (**C**), POD (**D**) and SOD (**E**) activity in the WT and *AtruLEA1*-overexpressing seedlings under normal and stress treatments. Significance was tested by using Student’s *t*-test (* *p* < 0.05 and ** *p* < 0.01).

**Table 1 plants-14-00117-t001:** Analysis data of protein physicochemical properties.

Protein Parameters	AtruLEA1
Molecular formula	C _781_ H_1284_ N _242_ O _296_ S _4_
Number of amino acids	181
Molecular weight	18,928.47 kDa
Isoelectric points	8.60
Negative charged residues	25
Positive charge residues	27
Instability index	18.14
Total mean value of the hydrophilicity	−1.256

## Data Availability

Data are contained within the article and Appendix A.

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
