# Peer review of "Overexpression of *AtruLEA1* from *Acer truncatum* Bunge Enhanced *Arabidopsis* Drought and Salt Tolerance by Improving ROS-Scavenging Capability"

_plants, 2025, doi:10.3390/plants14010117_

Round 1
Reviewer 1 Report
Comments and Suggestions for Authors
The article titled "Overexpression of AtruLEA1 from Acer truncatum Bunge Enhanced Arabidopsis Drought and Salt Tolerance by Improving ROS Scavenging Capability" focuses on studying some functional properties of the late embryonic abundant (LEA) gene from Acer truncatum Bunge. It is important to note that the general properties of proteins in this family have been extensively studied, and this work does not present any fundamental discoveries. The authors appear to emphasize the specific origin of the AtruLEA1 gene rather than offering novel insights into its function. It remains unclear what practical advantages the use of this particular gene provides. Moreover, if the study is aimed at practical applications, the choice of Arabidopsis as a model organism, which lacks economic significance, is questionable.
Another significant drawback is the frequent references to transcriptional studies of Acer truncatum that supposedly justify the selection of this gene. However, the manuscript neither provides a citation for this work nor includes original data. Additionally, there is no accession number for this gene in GenBank or other publicly available databases, raising concerns about the availability and reproducibility of the data.
Given the lack of significant scientific novelty and practical relevance, coupled with the overall low quality of both the research and its presentation, I recommend rejecting this manuscript. Below are some specific comments:
Line 13: The first sentence is missing a noun; it should be either "proteins" or "genes."
Line 17: The genus name should be abbreviated upon second mention.
Line 20: It is necessary to specify in which plant the AtruLEA1 gene was overexpressed.
Line 35 and throughout the text: References are cited without brackets, making the text difficult to follow.
Line 37: The abbreviation "LEA" for "late embryonic abundant" should be introduced at first mention, with subsequent use of the abbreviation only.
Line 41: ABA and ethylene are not abiotic stresses in themselves.
Line 51-52: The meaning of the sentence is unclear—what relevance do bananas have?
Line 53-55: Disconnected sentences lacking coherence.
Line 57: Gene names should be italicized.
Line 58-66: The paragraph is incoherent—specific groups of LEA proteins and their properties need to be clearly identified.
Line 69: The mechanism referenced is unclear.
Line 72: A reference for the cited study is missing.
Line 87: The correct spelling is "TAIR."
Figure 1: The electrophoresis of the AtruLEA1 PCR product is not informative and should be removed. The phylogenetic tree is of poor quality; bootstrap values below 50 should be omitted, as they are not statistically significant. The significance of the motif picture is unclear—if the authors intend to show that AtruLEA1 contains important domains, they should provide an alignment with marked amino acid sequences and explain their relevance in the text with references to original research.
Figure 2: The figure quality is very low, making it difficult to interpret. The relationship between the domains indicated and the conserved domains in Figure 1 is unclear. The term "protein low complexity" requires clarification.
Line 111: Specify which proteins are being discussed.
Chapters 2.1-2.6: These sections offer only superficial in silico characterizations of the AtruLEA1 protein, with limited scientific value. These predictive models could serve as a starting point for research but do not constitute research themselves. The content could be condensed into a short chapter, with only relevant figures included in the main body of the article.
Line 158: Expand the abbreviation "OE."
Line 158: Clarify how transgenic plant lines were "obtained via PCR."
Line 162: The content in brackets is unclear and needs clarification.
Line 163: "Mannitol" should be written with a lowercase "m."
Line 171: Ensure consistency in units—mass is given in milligrams above, but in grams here.
Figure 6: The quality of the image is poor, making it difficult to discern details. Why are there no error bars in the mean values on Figure 6A?
Figure 8B: The error in the mean value of WT on ABA suggests that the differences in transgenic plants may not be statistically significant.
Lines 257-258: The names of the enzymes should be capitalized.
Line 288: A reference to the transcriptome study of Acer truncatum is missing.
Line 290: The function of At3G15670.1 and the significance of its similarity to AtruLEA1 need to be explained.
Line 300: The statement is inaccurate—Figure 7D shows that RWC in WT and transgenic plants is generally the same.
Lines 301-312: No references are provided to support the claims. Additionally, the mere presence of domains does not confirm their functionality.
Line 383: The information presented here is not relevant to this chapter. The transcriptomic results should be cited from published data or be presented in this article.
Line 391: There is a reference to supplementary materials (Table S1), but these were not included in the submission.
Overall, the materials and methods, especially regarding molecular studies, are described very briefly and lack accuracy.
Comments on the Quality of English LanguageThe overall quality of the language in this manuscript is poor, which significantly detracts from the clarity and readability of the work. Numerous grammatical errors, awkward phrasing, and inconsistent terminology make it difficult to follow the authors' arguments and understand the key findings. Additionally, the lack of proper sentence structure and coherence in many sections leads to confusion and hinders the effective communication of the research.
Author Response
Line 13: The first sentence is missing a noun; it should be either "proteins" or "genes."
Response:It's a gene, already italicized
Line 17: The genus name should be abbreviated upon second mention.
Response:It has been abbreviated.
Line 20: It is necessary to specify in which plant the AtruLEA1 gene was overexpressed.
Response: The gene was overexpressed in Arabidopsis.
Line 35 and throughout the text: References are cited without brackets, making the text difficult to follow.
Response: Brackets was be added.
Line 37: The abbreviation "LEA" for "late embryonic abundant" should be introduced at first mention, with subsequent use of the abbreviation only.
Response: It has been changed.
Line 41: ABA and ethylene are not abiotic stresses in themselves.
Response: they were deleted.
Line 51-52: The meaning of the sentence is unclear—what relevance do bananas have?
Response: it has been changed.
Line 53-55: Disconnected sentences lacking coherence.
Response: And is added.
Line 57: Gene names should be italicized.
Response: it has been changed.
Line 58-66: The paragraph is incoherent—specific groups of LEA proteins and their properties need to be clearly identified.
Response: It has been changed.
Line 69: The mechanism referenced is unclear.
Response: it has been changed.
Line 72: A reference for the cited study is missing.
Response: the references were added.
Line 87: The correct spelling is "TAIR."
Response: it has been changed.
Figure 1: The electrophoresis of the AtruLEA1 PCR product is not informative and should be removed. The phylogenetic tree is of poor quality; bootstrap values below 50 should be omitted, as they are not statistically significant. The significance of the motif picture is unclear—if the authors intend to show that AtruLEA1 contains important domains, they should provide an alignment with marked amino acid sequences and explain their relevance in the text with references to original research.
Respones:
Figure 2: The figure quality is very low, making it difficult to interpret. The relationship between the domains indicated and the conserved domains in Figure 1 is unclear. The term "protein low complexity" requires clarification.
Response: Figure 2 has been replaced with a clearer image
Line 111: Specify which proteins are being discussed.
Response:
Chapters 2.1-2.6: These sections offer only superficial in silico characterizations of the AtruLEA1 protein, with limited scientific value. These predictive models could serve as a starting point for research but do not constitute research themselves. The content could be condensed into a short chapter, with only relevant figures included in the main body of the article.
Response: These studies are necessary to better investigate the gene function of LEA.
Line 158: Expand the abbreviation "OE."
Response: it has been expanded.
Line 158: Clarify how transgenic plant lines were "obtained via PCR."
Response: it should be “obtained via qRT-PCR”
Line 162: The content in brackets is unclear and needs clarification.
Response: It has been changed.
Line 163: "Mannitol" should be written with a lowercase "m."
Response: it has been changed to “mannitol”.
Line 171: Ensure consistency in units—mass is given in milligrams above, but in grams here.
Response: The unit has been modified to be consistent.
Figure 6: The quality of the image is poor, making it difficult to discern details. Why are there no error bars in the mean values on Figure 6A?
Response: It has been changed.
Figure 8B: The error in the mean value of WT on ABA suggests that the differences in transgenic plants may not be statistically significant.
Response: This result could indicate certain issues and should not be removed.
Lines 257-258: The names of the enzymes should be capitalized.
Response: The names of the enzymes have been capitalized.
Line 288: A reference to the transcriptome study of Acer truncatum is missing.
Response: The reference has been added.
Line 290: The function of At3G15670.1 and the significance of its similarity to AtruLEA1 need to be explained.
Response: This is a possible conclusion drawn from the evolutionary tree.
Line 300: The statement is inaccurate—Figure 7D shows that RWC in WT and transgenic plants is generally the same.
Response: RWC in WT and transgenic plants is generally the same in normally.
Lines 301-312: No references are provided to support the claims. Additionally, the mere presence of domains does not confirm their functionality.
Response: The references were added.
Line 383: The information presented here is not relevant to this chapter. The transcriptomic results should be cited from published data or be presented in this article.
Response: The reference was added.
Line 391: There is a reference to supplementary materials (Table S1), but these were not included in the submission.
Response: The table S1 was added.
Reviewer 2 Report
Comments and Suggestions for Authors
Summary
Using gene over-expression analysis, the authors functionally characterized LEA1 from Acer truncatum in Arabidopsis to understand the gene's role in abiotic stress tolerance. They found that in comparison with wild type plants, over-expression Arabidopsis lines showed enhanced tolerance to drought and salt stress conditions. In addition, the authors provided some insights to possible mechanisms through which the afore-mentioned tolerances were achieved. Overall, the study has good value in unraveling the specific role of AtruLEA1 in plant stress tolerance.
General Concept Comments
The authors need to explain why t-test (which usually compares 2 sample means) was chosen as the appropriate test-statistic to compare 3 samples (WT, OE3 and OE10) in Figs 6C&D instead of analysis of variance (ANOVA). Also, the number of plants analyzed in each biological replicate in Figs. 6, 7 and 8 was not clearly stated. Information on how replication was done should be stated.
Specific comments
Figures (such as Figs. 2, 6B ) had very poor quality and thus difficult to understand and match with the associated graphs. Figure quality has to be improved significantly.
Line 459: What exactly were compared with "t-test"?
Line 410: What does "DNA technology" mean?
Line 373: What does M. metabolica mean?
Comments on the Quality of English LanguageThe entire manuscript is replete with grammatical errors, misspellings, poor constructions, misuse of present and past tense, etc. This manuscript must be passed through a qualified English language editor who would diligently go through line by line and improve the language significantly.
Author Response
Figures (such as Figs. 2, 6B ) had very poor quality and thus difficult to understand and match with the associated graphs. Figure quality has to be improved significantly.
Response: The figure 2 was changed.
Line 459: What exactly were compared with "t-test"?
Response: It has been revised.
Line 410: What does "DNA technology" mean?
Response: it mean “qRT-PCR”, and it was revised.
Line 373: What does M. metabolica mean?
Response: It has been revised.
Reviewer 3 Report
Comments and Suggestions for Authors
This study focus on the AtruLEA1 gene derived from Acer truncatum Bunge, which exhibits significant potential in enhancing tolerance to both salt and drought stress in Arabidopsis thaliana. By modulating ROS levels, the gene plays a pivotal role in protecting plant cells from oxidative damage during abiotic stress conditions. Overall, I think this manuscript should be accepted after minor revision. but I only have a few minor concerns:
1. There are many other transcription factors involved in abiotic stress responses. Therefore, it is suggested that this article briefly introduce the research of other transcription factors in abiotic stress.
2. Figure 2: It is recommended to improve the clarity.
3. In the subcellular localization prediction section, it is recommended to add relevant experimental validation.
4. The fresh weight of the overexpressed lines under salt stress in Figure 6C exceeds that of the non-stressed line, which is questionable.
5. Fig. 8A: The segmentation between different lines should be more obvious.
6. Why some physiological indexes are opposite to the performance of two transgenic lines under salt stress?
Author Response
- There are many other transcription factors involved in abiotic stress responses. Therefore, it is suggested that this article briefly introduce the research of other transcription factors in abiotic stress.
Response: We would make corresponding considerations.
- Figure 2: It is recommended to improve the clarity.
Response: It was replaced.
- In the subcellular localization prediction section, it is recommended to add relevant experimental validation.
Response: Further experiments will be conducted in the future.
- The fresh weight of the overexpressed lines under salt stress in Figure 6C exceeds that of the non-stressed line, which is questionable.
Response: It has been revised.
- 8A: The segmentation between different lines should be more obvious.
Response: It has been revised.
- Why some physiological indexes are opposite to the performance of two transgenic lines under salt stress?
Response: There are differences in biological repetition.
Round 2
Reviewer 1 Report
Comments and Suggestions for Authors
After a second thorough review, I must inform you that the critical issues raised during the initial round of review remain largely unresolved.
As mentioned previously, the study on AtruLEA1 from Acer truncatum does not introduce any novel insights into the functional properties of LEA proteins, a well-studied family. While the specific origin of the gene is emphasized, this does not compensate for the lack of fundamental discoveries or practical relevance. You might consider submitting the manuscript to a another journal that aligns more closely with the scope of your research.
A significant portion of your manuscript consists of basic bioinformatic analyses that lacks the depth and novelty expected for a research article.
Another serious concern remains the lack of citation for the transcriptional studies of Acer truncatum that supposedly support the gene selection. Despite your response to the first review, no reference or original data has been provided to justify this, and the absence of a GenBank accession number for the AtruLEA1 gene further compromises the reproducibility and transparency of your work.
In conclusion, due to the persisting issues regarding scientific novelty, practical relevance, and overall manuscript quality, I must maintain my recommendation to reject the manuscript.
Comments on the Quality of English LanguageThe quality of the English in the manuscript is quite poor and significantly affects the readability and clarity of the work. There are numerous grammatical errors, awkward sentence structures, and inconsistent use of terminology throughout the text.
Author Response
Comments:As mentioned previously, the study on AtruLEA1 from Acer truncatum does not introduce any novel insights into the functional properties of LEA proteins, a well-studied family. While the specific origin of the gene is emphasized, this does not compensate for the lack of fundamental discoveries or practical relevance. You might consider submitting the manuscript to a another journal that aligns more closely with the scope of your research.
A significant portion of your manuscript consists of basic bioinformatic analyses that lacks the depth and novelty expected for a research article.
Another serious concern remains the lack of citation for the transcriptional studies of Acer truncatum that supposedly support the gene selection. Despite your response to the first review, no reference or original data has been provided to justify this, and the absence of a GenBank accession number for the AtruLEA1 gene further compromises the reproducibility and transparency of your work.
Response:The biological information of AtruLEA1 gene could be verified by bioinformatics, which can show certain function of AtruLEA1 gene.
We added previous articles as references or raw data to prove this, and adding the AtruLEA1 gene registry number increased the reproducibility and transparency of the work.

Round 3
Reviewer 1 Report
Comments and Suggestions for Authors
Dear Authors,
The manuscript now includes a reference to a transcriptome study on the effect of drought on Acer truncatum, and the AtruLEA1 number is also provided. Unfortunately, the study you referenced does not discuss AtruLEA1 or provide any information on how its expression changes in response to abiotic stress, making the rationale for selecting this gene still unclear. Instead of focusing on insignificant in silico analyses of AtruLEA1 protein parameters, it would have been more beneficial for the manuscript to leverage the data from the transcriptome study by dedicating a more detailed analysis to AtruLEA1, similar to what was previously done for AtruNAC36. This approach would have offered a stronger scientific foundation for the experimental work.
Additionally, the data presented in chapters 2.1–2.6 still lack scientific value and remain preliminary. These sections offer only basic bioinformatic assessments, which do not contribute significantly to the understanding of AtruLEA1's role in plant stress responses. As such, these findings do not meet the standard for publication in a scientific journal.
Given these unresolved issues, I cannot recommend the manuscript for publication.
Comments on the Quality of English LanguageThe overall quality of the manuscript's language is poor, which significantly impacts the clarity and flow of the content. The text contains numerous grammatical errors, awkward phrasing, and unclear sentence structures, making it difficult to follow the arguments and understand the scientific findings. In addition, inconsistencies in terminology and formatting detract from the readability of the paper. A thorough review and revision of the language are strongly recommended to improve the manuscript's readability and ensure clear communication of the research.
Author Response
In Figure 6C, the error bars for the WT data are not visible, which could impact the statistical analysis results depending on the deviations in the WT data.
Reponse: We have made the corresponding modifications and the error bars were now clear in figure 6C.
In Figure 8B, please verify the statistical difference between WT and OE3 for 'ABA,' as the error bars for WT overlap with those for OE-3 in figure 8B .
Response: We have verified the statistical difference between WT and OE3 for 'ABA' and added an asterisk.
In Figure 9, error bars should be shown in the graphs.
Response: The error bars were now added in figure 9.
In Figure 10, please review the statistical difference between WT and OE-3 under drought conditions for POD and SOD, as there is overlap in the error bars.
Response: We have recalculated the statistical differences between WT and OE-3 under drought conditions for POD and SOD, and added asterisks in figure 10.
